# Temporal Trends in Acute Coronary Syndrome Mortality in Serbia in 2005–2019: An Age–Period–Cohort Analysis Using Data from the Serbian Acute Coronary Syndrome Registry (RAACS)

**DOI:** 10.3390/ijerph192114457

**Published:** 2022-11-04

**Authors:** Ana Vasić, Zorana Vasiljević, Nataša Mickovski-Katalina, Stefan Mandić-Rajčević, Ivan Soldatović

**Affiliations:** 1Faculty of Medicine, University of Belgrade, 11000 Belgrade, Serbia; 2Institute of Public Health of Serbia “Dr Milan Jovanovic Batut”, 11000 Belgrade, Serbia; 3Institute of Social Medicine, School of Public Health and Health Management, Faculty of Medicine, University of Belgrade, 11000 Belgrade, Serbia

**Keywords:** acute coronary syndrome, time factors, trends, registries, risk factors

## Abstract

Background: Cardiovascular diseases ranked first in terms of the number of deaths in Serbia in 2019, with 52,663 deaths. One fifth of those were from ischemic heart disease (IHD), and half of IHD deaths were from acute coronary syndrome (ACS). We present the ACS mortality time trend in Serbia during a 15-year period using the latest available data, excluding the COVID-19 pandemic. Methods: The data on patients who died of ACS in the period from 2005 to 2019 were obtained from the National Statistics Office and processed at the Department of Prevention and Control of Non-communicable Diseases of the Institute of Public Health of Serbia. Number of deaths, crude mortality rates (CR) and age-standardized mortality rates (ASR-E) for the European population were analyzed. Using joinpoint analysis, the time trend in terms of annual percentage change (APC) was analyzed for the female and male population aged 0 to 85+. Age–period–cohort modeling was used to estimate age, cohort and period effects in ACS mortality between 2005 and 2019 for age groups in the range 20 to 90. Results: From 2005 to 2019 there were 90,572 deaths from ACS: 54,202 in men (59.8%), 36,370 in women (40.2%). Over the last 15 years, the number of deaths significantly declined: 46.7% in men, 49.5% in women. The annual percentage change was significant: −4.4% in men, −5.8% in women. Expressed in terms of APC, for the full period, the highest significant decrease in deaths was seen in women aged 65–69, −8.5%, followed by −7.6% for women aged 50–54 and 70–74. In men, the highest decreases were recorded in the age group 50–54, −6.7%, and the age group 55–59, −5.7%. In all districts there was significant decline in deaths in terms of APC for the full period in both genders, except in Zlatibor, Kolubara and Morava, where increases were recorded. In addition, in Bor and Toplica almost no change was observed over the full period for both genders. Conclusions: In the last 15 years, mortality from ACS in Serbia declined in both genders. The reasons are found in better diagnostic and treatment through an organized network for management of ACS patients. However, there are districts where this decline was small and insignificant or was offset in recent years by an increase in deaths. In addition, there is space for improvement in the still-high mortality rates through primary prevention, which at the moment is not organized.

## 1. Introduction

Worldwide, ischemic heart disease (IHD) has been the major leading cause of premature mortality for more than two decades [1]. IHD was responsible for 9.14 million global deaths in 2019, 49.2% of all deaths from cardiovascular diseases (CVD) in 2019, with an age-adjusted rate of 118 per 100,000 [2]. Compared to 2010, the number of deaths increased by 19.4%, whereas the age-adjusted death rate decreased by 9.7% in both males and females, indicating that, on average, global increases in IHD were due to population growth and aging [2]. Looking at the period from 2005 to 2015, the number of deaths due to IHD increased by 16.6% to 8.9 million deaths, whereas age-standardized mortality rates (ASR) for IHD decreased by 12.8% [3].

Dramatic declines in IHD mortality are achieved by improved diagnostics and treatment: the development of coronary care units providing continuous ECG monitoring, closed-chest resuscitation and external defibrillation, and the development of coronary arteriography, left ventriculography, and balloon angioplasty followed by the insertion of bare-metal stents. Intravenous streptokinase and the addition of the antiplatelet drug aspirin, in addition to newer platelet inhibitors, further reduced mortality, as did long-term administration of angiotensin-converting enzyme inhibitors, beta-adrenergic blockers and aldosterone blockers [4,5,6,7]. Of utmost importance for successful treatment proved to be the organization of coronary care units and primary percutaneous coronary intervention networks [8].

A significant reduction in mortality is attributed to the identification of causal risk factors, as primary prevention for individuals at risk for CVD reduces incident CVD event rates [7]. However, in many parts of the world there is no organized primary prevention [9]. Reduction in IHD incidence was also shown to be possible through population-wide interventions, as they shift the Gaussian distribution of the risk factor [10]. As an illustration, an impressive decline in IHD mortality in Austria was attributed to a campaign of hypertension awareness along with regulated access to all levels of health care and a choice of free providers delivered by the Austrian health service [11,12]. Applied measures in primary and secondary prevention impacted the decline in both IHD and CVD mortality [4,5,6,7,13].

Serbia is a middle-income country in Southeast Europe with a population of approximately seven million inhabitants living in 25 districts, covered by a comprehensive universal health system with free access to health care services, although financial barriers to medical care might be faced by some vulnerable groups. Healthcare is organized at three levels: primary, secondary and tertiary. CVDs ranked first in terms of the number of deaths in Serbia in 2019, with a share of 51.8% or 52,663 cases, where 17.7% died of IHD. Among those dying of IHD, 50.9% or 4700 people died of acute coronary syndrome (ACS), resulting in an age-standardized mortality rate for the European population (ASR-E) of 44.7/100,000 [14]. In 2005, the number of people who died of ACS was 7947, corresponding to an ASR-E of 77/100,000.

According to initial data on ACS in Serbia from 2003 when the national hospital registry was established, there was a very high ACS in-hospital mortality rate (18%) with thrombolysis applied only in 24.5% of acute myocardial cases. The use of beta-adrenergic blockers, statins, aldosterone blockers, and platelet inhibitors has become more frequent compared to the initial recorded statistics in 2003 [15].

There is no study on ACS epidemiology at the national level that studied the ACS mortality trend in total and by district in the last 15 years to take into account the introduction of significant changes in the treatment of this condition, such as pharmacotherapy and reperfusion therapy. Additionally, in the last five years, IHD ASR rates have increased in certain parts of the world [16]. Regardless, from Serbia, there are only several works dealing with mortality from CHD and ACS in certain districts [17,18,19,20,21,22]. In this study, age–period–cohort analysis was performed and used to reveal changes in mortality that affected all age groups at the same time, changes in mortality across specific cohorts, and most importantly, how mortality rate varied with age regardless of period.

The aim of this study was to examine temporal trends in ACS mortality in Serbia in the period 2005–2019 and discuss the potential reasons underlying revealed trends. In addition, age–period–cohort analysis was performed for the first time on ACS data from Serbia.

## 2. Materials and Methods

Data sources: Data on patients who died from ACS in the period 2005 to 2019 were collected from the National Statistics Office and processed at the Department of Prevention and Control of Non-communicable Diseases of the Institute of Public Health of Serbia. Number of deaths, crude mortality rates (CR) and age-standardized mortality rates for the European population (Age Standardized Rate—Europe, ASR-E) were analyzed. In registering deaths from ACS in Serbia, the International Classification of Diseases (ICD-10) was used. The codes for acute myocardial infarction, recurrent myocardial infarction and unstable angina are I21, I22 and I20.0, respectively [14].

CR were calculated using Serbian population data from the official censuses in 2011 as the denominator. For intercensus years, resident population estimates were received from the State Statistical Office database. The number of persons who died of ACS in 2018 was the numerator for the given year. The standardized rates were obtained using the direct standardization method, with the population of Europe as the standard population.

For age–period–cohort analysis, the year 2005 was added to create a 15-year period. Since there was no multicenter registry at this time, we worked with mortality data from the year 2006.

The quality of mortality data from Serbia is rated as medium by the World Health Organization [23].

Statistical Analysis: The annual percentage change (APC) in the time trend for the male and female population ages 0–75+ was obtained using joinpoint analysis. APC in mortality rate in men and women was calculated with National Cancer Institute (NCI) software, available online at https://surveillance.cancer.gov/joinpoint/ (accessed on 20 September 2022) [24]. The time trend refers to ASR-E rates. Due to the 15-year timeframe and five-year periods, the maximum number of joinpoints was set to two in order to obtain the best fit for the data. The *p* value was calculated for each model—with zero, one, or two joinpoints—and this value was used to select the model. We reported APC for the full period and the intervening joinpoints.

First, APC in ASR-E was analyzed for the whole territory of Serbia, for ages 0 to 75+, and for males and females separately. Next, APC ASR-E was calculated for each age group, from 25 to 75+ in males and 35 to 75+ in females, as those were the age groups with one or more death cases recorded. The trend was analyzed separately for males and females.

The true number of joinpoints was verified with the Monte Carlo Permutation method with an overall significance level of 0.05 and 4499 randomly selected data sets. The maximum number of joinpoints tested was five in each analysis. The Grid Search method was selected [25].

The comparability test, using the method proposed by Kim et al., was run in order to test disparities in mortality trends per age and per gender [26]. The comparability test is used to assess if the two regression mean functions are identical (test of coincidence) or parallel (test of parallelism). Results are not shown for the subgroups aged < 35 years, because less than five cases of ACS deaths were recorded in each five-year period.

An age–period–cohort model was used to assess the associations of age, period, and birth cohort with ACS mortality [27]. These associations were described in terms of longitudinal age curve, period relative risk, cohort relative risk, and annual percentage change in mortality rate for each age group relative to the overall annual percentage change across the whole study period. This method used the number of death cases in specified age groups during specified calendar periods, and the average populations of these age groups in the specified periods were set as input values [28]. In our study, age–period–cohort analysis was performed for 14 consecutive five-year age groups (20 to 24, 25 to 29, …, 85+), and the same five-year intervals for calendar periods (2005 to 2009, 2010 to 2014, 2015 to 2019) and birth cohorts (1916 to 1920, 1921 to 1925, …, 1990 to 1994 and 1995 to 1999). In order to apply joinpoint analysis to the 15-year period and analyze the 5-year intervals with complete data for age groups and districts, we approximated the year 2005 by using data from the year 2006. The reason to choose 2006 data was that mortality in this period was stable, while mortality for ACS in 2020 was affected by COVID-19 (the estimation of its impact was not the aim of this work).

The period effects were calculated in order to reveal changes over time periods that affected all age groups simultaneously, arising from changes in social, cultural, economic, or physical environments.

Cohort effects were associated with changes across groups of individuals with the same birth years. Net drift represents the overall log-linear trend by period and birth cohort and indicates the overall annual percentage change in the expected age-adjusted rates over time. Local drift represents the log-linear trend by period and birth cohort for each age group and indicates the annual percentage change in the expected age-specific rates over time. The longitudinal age curve indicates the expected age-specific rate in a reference cohort adjusted for period effects [27].

The period relative risks are the ratio of age-specific rates in each period relative to the reference period. Cohort relative risks are the ratio of the age-specific rate in each cohort relative to a reference cohort.

Reference age, reference period and reference cohort were calculated as shown in Appendix A [29].

Age–period–cohort analysis was performed using additional NCI software, retrieved from http://analysistools.nci.nih.gov/apc (accessed on 20 September 2022) [28]. The Wald χ^2^ test was adopted to test the significance of the estimable parameters and functions. All statistical tests were two-tailed.

Diagrams describing the Methods are included as Appendix A.

## 3. Results

From 2005 to 2019 in Serbia there were 90,572 deaths from ACS: 54,202 in men (59.8%) and 36,370 in women (40.2%). In the last 15 years, the numbers of deaths significantly declined by 46.7% in men and 49.5% in women (Table 1). Annual ASR-E declined in both genders. Average annual ASR-E in this period was 77.9/100,000 for males and 36.5/100,000 for females.

Significant declining trends in ACS mortality were observed in both men and women. The APC in ACS deaths from 2005 to 2019 was −4.4% in men using two joinpoints, and −5.8% in women using no joinpoints (Figure 1). The relationship between ASR-E and the number of primary Percutaneous Coronary Interventions (pPCI) per million population is presented in Appendix A.

Among men, a non-significant decrease in ACS mortality from 2005 to 2007 (−0.9% per year) was followed by a significant decrease from 2007 to 2016 (−6.8% per year), then a non-significant increase from 2016 to 2019 (+0.6 per year). Among women there were no joinpoints in the observed period. Over the full period, the highest significant decline in deaths was seen in women aged 65–69 (APC = −8.5), followed by women aged 50–54 and 70–74 (APC = −7.6%). In men, the highest decline was recorded in the 50–54 age group (APC = −6.7) and the 55–59 age group (APC = −5.7%) (Table 2).

Looking at the full period, in the whole country significant declines in deaths among men and women were observed, more so for women (Table 3). However, from 2016 onwards, death rates increased in men, revealed by a non-significant joinpoint (Table 2).

Table 3 shows the APC for full period across the 25 districts of Serbia. In all districts there was a significant decline in deaths in both genders, except in Zlatibor, Kolubara and Morava, where increases were recorded. In addition, in Bor and Toplica almost no change was observed over the full period in both genders.

Figure 2 shows estimates of age effects on ACS mortality, in men and in women. The risk of death caused by ACS increases markedly with age in both genders. In men, the age effect curve has a sharp incline from just before 40 years of age to 55, is flat until 65, and drops by 75 years of age, followed by a steep rise after 75.

In women the mortality rate increases just after 40 years of age. The rate continues to grow following a similar pattern from 40 to 70 years of age, with the rate increase being highest after the early 70s (Figure 2).

Period effects show a decline in ACS mortality for both men and women, which is higher in women. Mortality decreased more in the first two periods analyzed than in the latest one, 2015–2019 (Figure 3.)

Cohort effects show progressive improvements in those born from 1925 and on. Improvements are of a similar pattern in men and women, with higher RR in women (up to 8.4) than in men (up to 6.7). In youngest cohort low number of cases is recorded, which resulted in wide CI (Figure 4).

Net drifts show striking mortality reduction in both genders, with overall annual change being higher in women, with −7.7% [CI95%: −9,4%, −6.1%] compared to −5.6% [CI 95% −6.6%,−4.6%] in men. Local drifts were very similar in men and women at all ages. The highest decrease was seen in men aged 30–34 and 60–64. In women, the highest APC was observed for the 30–34 and 65–69 age groups. A wide CI was seen in the youngest age group, due to the small number of cases (Figure 5).

## 4. Discussion

Over the past 15 years, there was a significant decline in ACS mortality in Serbia—from 2005 to 2019 the number of deaths reduced by 46.7% in men and 49.5% in women, giving for this period an average APC of 4.4% in men and 5.8% in women. At the same time, ASR-E reduced in both genders, even though the population of elderly people in Serbia increased, implying significant improvements in patient management. During this period, the incidence of ACS cases increased in terms of number, with the incidence of ASR-E remaining the same, again indicating improved patient care. In terms of age-specific mortality rates, the largest decline in Serbia was seen in age groups below 75, probably due to those patients being eligible for PCI: women aged 65–69 (APC = −8.5%), followed by women aged 50–54 and 70–74 (APC = −7.6%). In men, the best improvements were seen in the 50–54 (APC = −6.7%) and 55–59 (APC = −5.7%) age groups.

The trend of decreasing ACS mortality with a stable or increasing trend of incidence is an indicator of efficient treatment and secondary prevention. Had there been a decrease in incidence or stable incidence in the aging population, it would indicate efficient primary prevention. Primary CVD prevention in Serbia is not organized and is instead conducted on the individual level in primary health care centers, and is based on various forms of counseling for lifestyle modification and identification of high-risk patients using SCORE charts. There were many initiatives from cardiology societies: since 2002 Serbia actively participated in the Heart Failure Awareness Day, the “25 by 25” global campaign for a 25% reduction in CVD mortality by 2025, and the “Pace for Heart” campaign, which promotes physical activity for the prevention of cardiovascular diseases [9]. This could explain the decline in crude incidence rates in last 15 years in the 50–54, 60–64 and 65–69 age groups. However, health surveys in the general population in this period showed no improvements in cardiovascular risk factors; hypertension, dyslipidemia, diabetes mellitus, smoking and obesity prevalence were higher in 2019 compared to 2006 [30,31].

Improvements in the acute management of ACS in Serbia, as elsewhere, started with the opening of coronary care units (CCU), followed by the introduction of streptokinase and, later, thrombolytic drugs, the introduction of coronary artery catheterization, balloon angioplasty, and surgical revascularization. The use of emergency medical services helped to reduce the time between symptoms onset and intervention. 

The number of CCUs in Serbia increased from 49 in 2005 to 54 in 2008 and 56 in 2014. Thrombolysis was the only reperfusion strategy until 2005, and after that remained the dominant reperfusion strategy. Previous studies showed that the mortality with thrombolysis was 11.8%, decreasing to 6% if PCI was added and 4.1% if the STEMI patient was treated with pPCI [8,15,32]. The application of pPCI started during 2006. In 2009 in Serbia there were only seven PCI centers (24/7), most of them more than 50 km from the closest CCU. After Serbia joined the Stent for Life Initiative in 2009, the rate of pPCI treatment increased 2.3-fold, from 311 per million inhabitants in 2009 to 717 per million inhabitants in 2015. That was a 4.9-fold improvement compared to 2007. The emergency medical services’ (EMS) average time from being contacted to reaching the patient was 12.4 min in 2015 in the capital, Belgrade. Organizing non-PCI hospitals, ambulance, CCUs and PCI centers across Serbia in one network for the management of ACS patients evidently reduced mortality. The increased number of pPCIs improved patient survival [33]. In Serbia, however, the overall proportion of untimely reperfused eligible STEMI patients remains high. This might be caused by an insufficient PCI network or unused pharmaco-invasive therapies. Patients treated without reperfusion therapy have higher mortality (15.7%) than patients treated with fibrinolytic therapy (10.5%) or pPCI (6.2%). One of the significant predictors of the withdrawal of the application of any RT was shown to be the time to first medical contact (>360 min) [34]. Table 2 clearly reflects these improvements achieved in patient management. Looking at the trend in age-specific mortality rates, significant joinpoints in the most affected groups correspond to broad use of thrombolytic therapy and the Stent for Life initiative: in men 2007–2012 in the 60–64 age group (−10.4% [−13.8,−6.9]), 2009–2015 in the 65–69 age group (−9.6% [−13.4,−5.7]), 2009–2012 in the 70–74 age group (−11.5% [−24.3,3.4]); in women 2007–2011 in the 60–64 age group (−13.5% [−22.1,−4]) (Table 2).

A 30-year IHD mortality trends across Europe identified a significant mortality decline in developed Western European countries and small improvements in Southeastern and Eastern European countries, where mortality rates were several folds higher than in Western European countries, even though there was a period of a decreasing ACS trend from 2005 to 2013 in Eastern Europe [35]. Mortality data for IHD submitted to the WHO together with data on CVD risk factors from 2005 to 2015 revealed that mortality rates fell in most of the countries included in the study. The study concluded that the decrease in IHD mortality was not associated with a parallel decrease in the prevalence of cardiovascular risk factors, but rather was a result of better treatment of cardiovascular risk factors or an improvement in health care systems [35]. 

It is well reported in the literature that the ACS mortality reduction in recent decades was achieved through increases in the use of pharmacotherapy and interventional cardiology procedures, implementing specific guidelines from cardiology societies, and the development of organized networks for patient management that include emergency medical services [11,35,36,37,38,39,40,41]. In Europe, the lowest IHD mortality rate per 100,000 inhabitants, with the lowest and declining age-standardized rate relative to the WHO World Standard Population (ASR-W), is seen in France and Spain (ASR-W of approximately 20 per 100,000 in 2015), followed by Norway (approximately 30 per 100,000 in 2015) and Italy (approximately 40 per 100,000 in 2015) [35]. France was one of the first countries to use pharmacokinetic therapy. In addition, France has the third largest number of PCI hospitals in Europe, all with 24/7 service. Italy has the second largest number of PCI hospitals, with a favorable mean population per pPCI centre (24/7 service). Spain is a country with one of the lowest IHD mortality rates in the world, and this rate decreased despite significant population aging, mainly because of a decrease in deaths among patients who arrive at a hospital for treatment [36]. In Norway, the decline in CHD mortality was attributed to a decline in out-of-hospital deaths and reduced incidence due to improvement in some of the CHD risk factors. In addition, the number of STEMI cases decreased, while revascularization and the proportion of patients receiving medical therapy for secondary prevention at discharge increased over time [37]. In Germany, the country with the highest number of PCI hospitals in Europe, a decline in the ACS mortality rate was reported to be the result of an increased percentage of coronary angiographies and PCIs performed [42]. Switzerland, Germany and Poland had the highest numbers of PCIs per 100,000 inhabitants in 2010/2011. In Switzerland a large decline in the ACS mortality rate was assigned to increased rates of immediate medical therapy, and increased rates of coronary angiography and PCIs were reported to be the strongest independent predictors of survival [38]. The causes associated with a reduction in ACS mortality in Poland were a significant increase in pPCIs and, as in Qatar, the increased use of pharmacotherapy for medical treatment and secondary prevention [39,43]. 

In our study, over the full period the whole country showed a significant decrease in deaths in both genders, more so in women. However, from 2016 onwards, there was an increase in death rates in men, revealed by a non-significant joinpoint, and a similar increase in women, with no joinpoints. Looking at the locations where this was seen, it was found to occur in Morava, Toplica, Kolubara and Zlatibor districts. The initial severity of the illness could be one of the reasons for differences in *mortality* rates; patients living in rural areas are less likely to reach hospital within two hours of the onset of symptoms. Another problem at certain locations is that pPCI is available from Monday from 07 h to Friday at 13 h only; outside those hours, STEMI patients are treated with thrombolytic or classical therapy according to current recommendations [17,18]. Recent studies reported, besides the limited availability of PCI in the first shift, a limited number of trained interventional cardiologists and high downtime for Cath Lab machines [22]. The increased or unchanged mortality rates seen at several locations could be related to the organization of transport, which depends on how well the emergency services are equipped, or the quality of the roads [22]. A study that included all patients from the ACS registry with complete data treated over a three-year period, 2007–2009, revealed that pPCI was used for 56.7% of patients in centers that operate 24 h/7 d and 25.1% of cases in centers operating 8 h/5 d [44]. This study showed that patients were transported to the nearest health center more often by EMS than on their own. Nevertheless, it was seen that a great number of patients were sent to centers with no PCI available [44]. In addition, what affects time to intervention is not just the distance from the nearest coronary unit to the nearest PCI center, but the quality of the roads, which may require limited speed. The geographic terrain south of Belgrade, particularly in south Serbia, is somewhat challenging, especially in Toplica district.

This is the first work uncovering age–period effects in ACS mortality in Serbia. It reveals patterns that can be used in prevention and treatment by focusing on improving care for particular cohorts or age groups. In addition, it reveals the impact of any interventions applied in certain periods.

The longitudinal age curve in men shows that the risk of death caused by ACS increased markedly with age, with the highest mortality rate being much less than reported in other countries [35]. In women, the age effect shows exponential growth after 50 years of age and the steepest gradient after 70 years of age, with maximums similar to the USA and less than those in Eastern Europe, Brazil, or the UK [35].

Period effects show favorable results in both genders, with greater improvements in women, probably due to previous differences in presentation and disparities in treatment [45,46]. The highest reduction in mortality was seen in the first two periods: 2005–2009 and 2010–2014. The first period reflects the increased number of coronary care units, the introduction of streptokinase and the increased use of thrombolytic therapy. The second period showed improvements due to successful implementation of the Stent for Life initiative starting from 2009, together with the organization of coronary care units, pPCI networks and emergency medical systems [8,33]. 

Cohort effects in Serbia present striking improvements in those born after 1925 and progressively onwards to the latest cohorts followed (born after 1999). This is similar to reports elsewhere [27].

The strengths of this study are the national representation of ASR mortality over a 15-year period and the application of two methods (the joinpoint regression method and the age–period–cohort method) for analysis of the trend in ASR mortality. The APC trend is reported for all 25 districts in the country.

The limitations are related to the limitations of age–period–cohort analysis, such as collinearity between age, period and cohort effect. As this study is a descriptive epidemiological study, the temporal trend at the population level does not necessarily apply to individuals.

## 5. Conclusions

The mortality rate from acute coronary syndrome in Serbia over the last 15 years significantly declined, most probably due to advances in diagnostics and treatment, as evidence suggests that no primary prevention was implemented to affect this drop in mortality rates. After a period of large decline from 2005 to 2016, in line with tremendous progress in organized ACS diagnostics and therapy in Serbia, there was a slow increase in the number of deaths and ASR-E in men and women in Serbia, particularly in the eldest age group and in men, which means that there are factors beyond population aging influencing the mortality trend. Subpopulations within the country over the last four years experienced adverse trends in ACS mortality, suggesting a need to support the delivery of interventions in high-risk locations. In the last four years there was no improvement in mortality rates, indicating that the national health system will ultimately need to address organized ACS-related primary preventive services, particularly bearing in mind that the population is aging.

## Figures and Tables

**Figure 1 ijerph-19-14457-f001:**
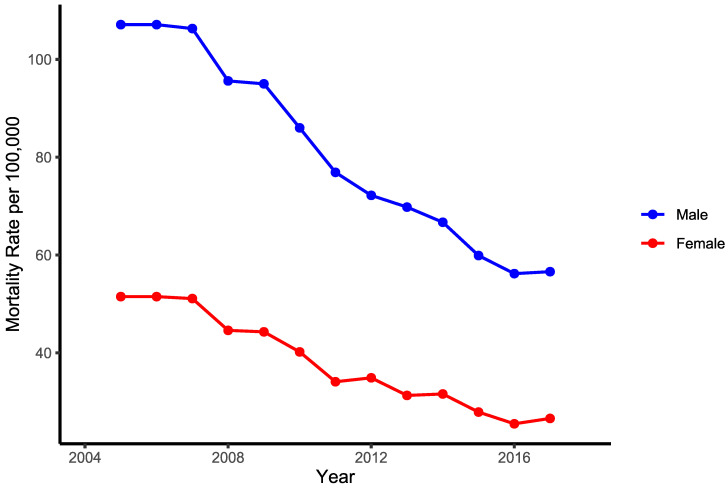
Joinpoint analysis: trends in age-specific mortality rates from acute coronary syndrome (per 100,000 persons) for males and females in Serbia, 2005–2019; Statistically significant trend. AAPC, average annual percent change; ASR, age-standardized rate (per 100,000 persons, using European standard population).

**Figure 2 ijerph-19-14457-f002:**
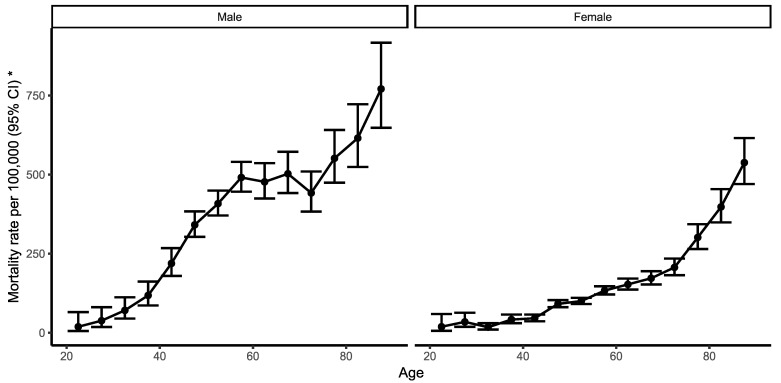
Age effects, Male (**left**) and Female (**right**). Longitudinal age curves are estimated via the age–period–cohort model and indicate the expected age-specific rate of ACS mortality per 100,000 in a reference cohort adjusted for period effects (marked by *).

**Figure 3 ijerph-19-14457-f003:**
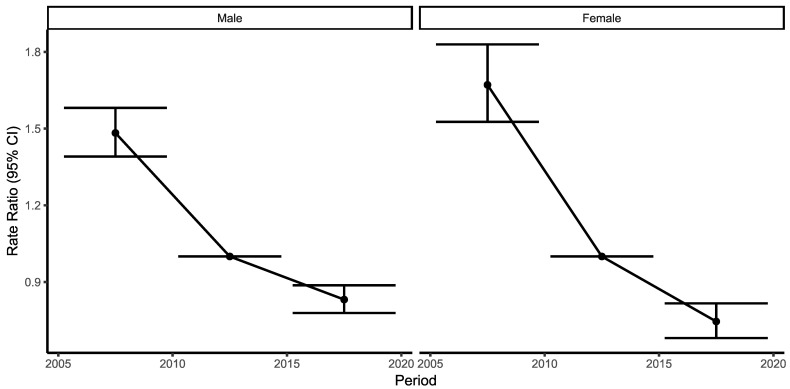
Period effects, Male (**left**) and Female (**right**). Relative risk of each period compared with the reference period—ratio of age-specific rates in each period relative to the reference period.

**Figure 4 ijerph-19-14457-f004:**
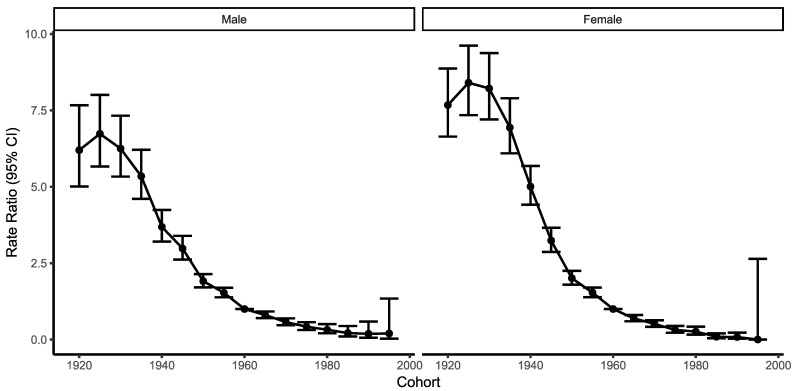
Cohort effects, Male (**left**) and Female (**right**). Cohort relative risk is the ratio of the age-specific rate in each cohort relative to a reference cohort. The wide CI in the youngest cohort is a result of the low number of cases.

**Figure 5 ijerph-19-14457-f005:**
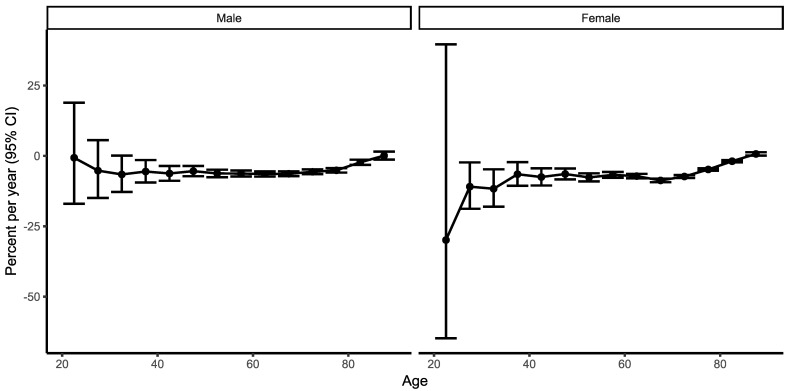
Local drifts with net drift, Male (**left**) and Female (**right**). Net drift indicates the overall annual percentage change across the whole study period. Local drift indicates the annual percentage changes in mortality rate for each age group relative to the net drift.

**Table 1 ijerph-19-14457-t001:** Number of deaths and ASR-E in men and women, 2005–2019.

	All		Males		Females	
Year	Number	ASR-E	Number	ASR-E	Number	ASR-E
2005	7947	77	4757	107.1	3190	51.5
2006	7947	77	4757	107.1	3190	51.5
2007	7947	77	4757	106.3	3190	51.1
2008	7158	68	4305	95.6	2853	44.6
2009	7158	67.6	4305	95	2853	44.3
2010	6600	61.3	3907	86	2663	40.2
2011	5787	53.8	3517	76.9	2270	34.1
2012	5817	52.1	3430	72.2	2387	34.9
2013	5496	49	3329	69.8	2167	31.3
2014	5383	47.8	3189	66.7	2194	31.6
2015	4852	42.7	2890	59.9	1962	27.9
2016	4534	39.7	2720	56.2	1814	25.5
2017	4624	40.4	2757	56.6	1867	26.6
2018	4652	40.4	2768	56.7	1884	26.4
2019	4700	40.4	2814	57.1	1886	26
Overall	90572	55.6	54202	77.9	36370	36.5

**Table 2 ijerph-19-14457-t002:** Joinpoint analysis: trends * in age-specific mortality rates from acute coronary syndrome (per 100,000 persons) for males and females in Serbia, 2005–2019.

Age †	Males			
	Period	APC (95% CI)	Period	APC (95% CI)
All ages	2005–2007	−0.9 (−7.6 to 6.3)		
	2007–2016	−6.8 * (−7.8 to −5.8)		
	2016–2019	0.6 (−5.8 to 7.4)		
	Full period	−4.4 *^‡^ (−5.9 to −2.9)	Full period	−5.8 *^‡^ (−6.5 to −5.1)
25–29	Full period	−2.1^‡^ (−5.4 to 1.4)		N/A
30–34	Full period	−6.5 *^‡^ (−8.9 to −4.1)		N/A
35–39	Full period	−3.6 *^‡^ (−5.9 to −1.4)	Full period	−7.5 *^‡^ (−10.1 to −4.9)
40–44	Full period	−5.9 ^‡^ (−7.6 to −4.2)	Full period	−5.7^‡^ (−10.6 to −0.7)
45–49	2005–2008	1.5 (−7.9 to 11.8)		
	2008–2019	−6.4 * (−8.1 to −4.7)		
	Full period	−4.8 *^‡^ (−6.9 to −2.6)	Full period	−5.3 *^‡^ (−6.9 to −3.6)
50–54	Full period	−6.7 *^‡^ (−7.8 to −5.6)	Full period	−7.6 *^‡^ (−9.2 to −5.9)
55–59	Full period	−5.7 *^‡^ (−6.6 to −4.8)	Full period	−6.7 *^‡^ (−7.8 to −5.5)
60–64	2005–2007	−1.1 (−9.8 to 8.4)	2005−2007	−0.4 (−14.6 to 16.1)
	2007–2012	−10.4 * (−13.8 to −6.9)	2007−2011	−13.5 * (−22.1 to −4)
	2012–2019	−2.3 * (−4.6 to 0)	2011−2019	−3.5 * (−7 to 0)
	Full period	−5.1 *^‡^ (−6.9 to −3.4)	Full period	−6.1 *^‡^ (−9.3 to −2.7)
65–69	2005–2009	−2.1 (−6.3 to 2.4)		
	2009–2015	−9.6 * (−13.4 to −5.7)		
	2015–2019	−0.7 (−8.6 to 8)		
	Full period	−5 *^‡^ (−7.5 to −2.4)	Full period	−8.5 *^‡^ (−9.8 to −7.2)
70–74	2005–2009	−3 (−6.6 to 0.8)		
	2009–2012	−11.5 (−24.3 to 3.4)		
	2012–2019	−4 * (−6.6 to −1.2)		
	Full period	−5.4 *^‡^ (−8.3 to −2.3)	Full period	−7.6 *^‡^ (−8.8 to −6.4)
75+	2005–2017	−4.4 * (−5 to −3.8)		
	2017–2019	2.5 (−10.3 to 17.2)		
	Full period	−3.4 *^‡^ (−5.1 to −1.8)	Full period	−3.5 *^‡^ (−4.1 to −2.9)

* Statistically significant trend at the alpha = 0.05 level; † Joinpoint results are not shown for subgroups with fewer than five cases of ACS in any year; ‡ Average annual percent change presented across the full period, AAPC.

**Table 3 ijerph-19-14457-t003:** Annual percent change for full period 2005–2019 in 25 districts in Serbia.

All	Males	Females
	APC (95% CI)	APC (95% CI)
Serbia	−4.4 * (−5.9 to −2.9 )	−5.8 * (−6.5 to −5.1)
Belgrade	−5.2 * (−6 to −4.3 )	−4.1 * (−7.1 to −1)
West Bačka District	−6.1 * (−7.5 to −4.8)	−7.2 * (−9.0 to −5.4)
South Banat District	−8.9 * (−10.8 to −7.1)	−6.7 * (−8.4 to −5.0)
South Bačka District	−6.9 * (−9.8 to −4)	−8.8 * (−10.3 to −7.2)
North Banat District	−6.0 * (−9.5 to −2.3)	−2.7 (−7.9 to 2.7)
North Bačka District	−5.9 * (−9.6 to 2.0)	−5.8 * (−7.4 to −4.2)
Middle Banat District	−9.2 * (−11.3 to −7.2)	−10.0 * (−11.8 to −8.1)
Srem District	−9.3 * (−11.4 to −7.2)	−8.3 * (−10.2 to −6.4)
Zlatibor District	2.2 (0.0 to 4.4)	0.9 (−1.5 to 3.5)
Kolubara	1.2 (−2.4 to 4.9)	0.8 (−2.1 to 3.9)
Mačva District	−6.5 * (−8.3 to −4.6)	−7.7 * (−9.9 to −5.5)
Moravica District	−5.9 * (−7.8 to −4.1)	−6.7 * (−9.2 to −4.2)
Morava District	0.4 (−4.3 to 5.3)	3.0 (−3.1 to 9.5)
Rasina	−5.3 * (−6.6 to −4.0)	−7.0 * (−9.1 to −4.9)
Raška	−3.7 * (−4.6 to −2.7)	−5.0 * (−7.5 to −2.4)
Šumadija	−5.6 * (−10.0 to −0.9)	−6.5 * (−10.4 to −2.3)
Bor	−0.8 (−3.4 to 1.8)	−0.8 (−3.3 to 1.7)
Braničevo	−6.3 * (−8.7 to −3.9)	−6.9 * (−9.9 to −3.8)
Zaječar	−3.1 * (−4.2 to −2.0)	−6.2 * (−7.7 to −4.6)
Jablanica	−1.9 * (−3.5 to −0.2)	−3.4 * (−5.3 to −1.4)
Nišava	−2.7 * (−4.1 to −1.3)	−2.3 * (−4.0 to −0.6)
Pirot	−4.1 * (−5.6 to −2.6)	−8.5 * (−10.8 to −6.2)
Danube	−7.6 * (−10.1 to −5.1)	−8.2 * (−10.2 to −6.1)
Pčinj	−6.5 * (−8.2 to −4.8)	−6.2 * (−10.1 to −2.3)
Toplica	−2.1 (−7.1 to 3.2)	−0.9 (−4.2 to 2.5)

* Statistically significant trend at the alpha = 0.05 level.

## Data Availability

Not applicable.

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
