# Peer review of "Temporal Trends in Acute Coronary Syndrome Mortality in Serbia in 2005–2019: An Age–Period–Cohort Analysis Using Data from the Serbian Acute Coronary Syndrome Registry (RAACS)"

_ijerph, 2022, doi:10.3390/ijerph192114457_

Round 1

Reviewer 1 Report

ijerph-1968486

Thank you for the opportunity to review this paper. The manuscript is very well written and addresses ACS from a contemporary point of view. The design of the study, methods, results and especially discussion, are noteworthy.

Below you can find my comments, which are made with the desire to contribute and improve the final version of the manuscript:

1.     English – minor spelling errors

2.     “Yet, in many parts of the world there is no organized primary prevention (9).” – please provide an example of the possible and/or efficient primary prevention actions/goals

3.     The readers would benefit from a flowchart explaining the Methods section

4.     For the readers who are not familiar with Age-period-cohort, I suggest you provide an explanation of this method

Author Response

Answer can be found in the attached document

Reviewer 2 Report

The authors well described an important and relevant epidemiological situation in Serbia: mortality for acute coronary syndrome. The data was well exposed and the results are very relevant. However, the discussion part needs to be significantly improved.

The emergency network is briefly described in discussion. What about telemedicine service?

It would be useful to include an image that represents the emergency network in Serbia.

The number of PCI in Serbia is increasing by years. It would be useful to provide an image comparing the reduction in total mortality (males and females) with the opening of CCUs and the increase in the number of PCI.

I find the deductions inserted between lines 294 and 300 scientifically incorrect. Furthermore, this is not the place to talk about politics with reference to communism and capitalism. Lines 294-300 need to be deleted.

In the discussions, from line 316-334, I would avoid reviewing the medical therapy undertaken in the different European countries which, however, have been ubiquitously implemented through the Guidelines of the European Society of Cardiology which can still be cited in the bibliography. Generally it can be indicated that the reduction in mortality in Western European countries has improved over time thanks to the adoption of specific guidelines that have been implemented over the years by the European Society of Cardiology (Ref. Jean-Philippe Collet, Holger Thiele, Emanuele Barbato, Olivier Barthélémy, Johann Bauersachs, Deepak L Bhatt et al. 2020 ESC Guidelines for the management of acute coronary syndromes in patients presenting without persistent ST-segment elevation: The Task Force for the management of acute coronary syndromes in patients presenting without persistent ST-segment elevation of the European Society of Cardiology (ESC). European Heart Journal, Volume 42, Issue 14, 7 April 2021, Pages 1289–1367. Borja Ibanez, Stefan James, Stefan Agewall, Manuel J Antunes, Chiara Bucciarelli-Ducci, Héctor Bueno et al. 2017 ESC Guidelines for the management of acute myocardial infarction in patients presenting with ST-segment elevation: The Task Force for the management of acute myocardial infarction in patients presenting with ST-segment elevation of the European Society of Cardiology (ESC). European Heart Journal, Volume 39, Issue 2, 07 January 2018, Pages 119–177.)

In the graph of figure 1 it is necessary to specify that the Mortality Rate is calculated on 100,000 people; as in the graph of figure 2 it should be specified that the mortality rate is to be considered on the reference cohort adjusted for period effects

Formatting errors were detected in the text and need to be corrected: at line 76 de-veloped; at line 266 an-gioplasty; at line 267 revasculariza¬tion and emer¬gency; at line 268 re¬duce; to be removed the first point at line 276; at line 276 inhabit-ants; at line 282 pro-portion; at line 284 Pa-tients. Check for further formatting errors.

The bibliography needs to be double checked, some are incomplete or not in English, this makes it impossible for the reviewer and the reader to verify.

Author Response

The answers are available in the attached file.

Reviewer 3 Report

I applaud your scientific contribution through this manuscript.

it can fill the literature gap

I highly recommend choosing your article keywords by visiting:

 MeSH (Medical Subject Headings).

I have noticed using unnecessary hyphen. For example 

aspi-rin (page 2,line51)

It would be helpful highlighting your reader about the health care system in Serbia.

Author Response

The answers can be found in the attached document

Round 2

Reviewer 2 Report

The authors answered all the questions.

The work is excellent and certainly useful.

I only recommend checking the formatting of the text and making sure there are no colored or highlighted words.

Author Response

We thank the reviewer for the kind words and again for all the comments and suggestions which have helped improve our Manuscript.

We have checked the document and removed all the highlighted text and colored words.